# `DoStoVoQ`: Doubly Stochastic Voronoi Vector Quantization SGD for Federated Learning

## Abstract

The growing size of models and datasets have made distributed implementation of stochastic gradient descent (SGD) an active field of research. However the high bandwidth cost of communicating gradient updates between nodes remains a bottleneck; lossy compression is a way to alleviate this problem. We propose a new *unbiased* Vector Quantizer (VQ), named `StoVoQ`, to perform gradient quantization. This approach relies on introducing randomness within the quantization process, that is based on the use of unitarily invariant random codebooks and on a straightforward bias compensation method. The distortion of `StoVoQ` significantly improves upon existing quantization algorithms. Next, we explain how to combine this quantization scheme within a Federated Learning framework for complex high-dimensional model (dimension $> 10^6$), introducing `DoStoVoQ`. We provide theoretical guarantees on the quadratic error and (absence of) bias of the compressor, that allow to leverage strong theoretical results of convergence, e.g., with heterogeneous workers or variance reduction. Finally, we show that training on convex and non-convex deep learning problems, our method leads to significant reduction of bandwidth use while preserving model accuracy.

## 1 Introduction

In this paper, we consider the Federated Learning framework, in which a potentially large number $K$ of *workers* cooperate to solve the following problem:

$$\min_{\theta \in \mathbb{R}^D} \sum_{k=1}^{K} f_k(\theta), \tag{1}$$

where each function $f_k : \mathbb{R}^D \to \mathbb{R}$ represents the empirical risk on worker $k \in [K]$ (where $[K] = \{1, \ldots, K\}$) and $D$ is the ambient dimension of our problem. Each worker potentially holds a fraction of the data, and can share information with a central server, which progressively aggregates and updates the model accordingly [18, 17].

Stochastic gradient algorithms [28] are particularly well suited in the *large scale learning* setting [6, 7]. The methods can easily be adapted to the distributed (and more generally federated) learning framework; see [17] and the references therein. For synchronous distributed Stochastic Gradient Descent, at every iteration, given the current parameter $\theta_t$, each worker computes an unbiased estimate $g_{k,t+1}(\theta_t)$ of the gradient of the local loss function $f_k$. The central server then aggregates those oracles and performs the update.

Communicating the gradients from the local workers to the central server is often a major bottleneck. The drastic increase both in the number of parameters and of workers over the last years, has made this problem even more acute. Alleviating the communication cost is one of the crucial challenges of

federated learning [17, Sec. 3.5]. A central idea to tackle this issue is *communication compression*, which consists in applying a lossy compression to the parameters or gradients to be transmitted. Since compression alters the message transmitted, the number of iterations required to reach a given accuracy may increase, therefore compression is of interest in situations where the communication gains are large relative to the increase of communication rounds. The design of new compression schemes (see among others [30, 2, 4, 5, 34]) and the adaptation of the learning algorithms to this setting (see e.g. [32, 1, 35, 33, 36, 22, 26, 12, 11, 21] and the references therein) are an extremely active field of research.

Our main contribution is to introduce a novel **unbiased vector quantization** procedure allowing to reach **high-compression rate**, with a **small computational** overhead. More precisely, our contributions are as follow: first, we introduce `StoVoQ`, a vector quantization algorithm based on unitarily invariant random codebooks to automatically obtain **directionally unbiased** gradient oracles, and introduce a scalar **correction function**, that makes compression operator **unbiased** for a very modest computational cost. We further provide theoretical guarantees on the distortion of the compressor. In summary, `StoVoQ` algorithm is based on the following points, that are developed in Section 2.

1. **Vector quantization** The input vector $x \in \mathbb{R}^d$ is mapped onto its nearest neighbor in a codebook $\mathscr{C}_M = \{c_i\}_{i=1}^M$.
2. **Random codebook.** A **new codebook** is sampled every time a new quantization operation is performed. The proposed approach is different from classical random VQ which typically uses a random codebook, but which is sampled once and then kept fixed.
3. **Bias removal.** By relying on unitarily invariant distribution for the codewords generation, the quantized value of each vector $x \in \mathbb{R}^d$ is **directionnally unbiased**. The bias only depends on the number and distributions of the random of codewords and on $\|x\|$. This key property allows to derive a simple way to remove the quantization bias.

Then, we describe how to use `StoVoQ` within the FL framework: this yields the algorithm `DoStoVoQ`. We prove that this process satisfies a strong assumption on the compression process, that allows to automatically derive fast convergence rates. In Section 3, we describe `DoStoVoQ`, i.e., how we solve the optimization problem (1) in dimension $D$.

4. **Splitting and renormalizing gradients.** First, we split each gradient to compress into *buckets* $(x_i)_{i=1,...,L}$ of dimension $\mathbb{R}^d$, to use `StoVoQ` for each bucket.
5. **Synchronisation of random sequences of codebooks.** We ensure that those codebooks are independent, at each step and between each machine, by generating a new codebook each time. To avoid any subsequent communication cost, we synchronously generate the codebooks on the central and local servers, by initially sharing random seeds.

Remark that point 1 was also used in Dai et al. [8]. Points 2 to 3 and 5 are novel ideas that have not been leveraged in the FL framework. Finally, we demonstrate the effectiveness of random codebook quantization for gradient compression by extensive experiments in Section 4 on standard benchmarks like ImageNet or CIFAR10.

# 2 `StoVoQ` algorithm

Several compression operators [34, 27, 10, 4, 8, 36, 37] have been introduced recently as bandwidth reduction for distributed learning became a major challenge. In this section, we first discuss the importance of unbiasedness of compression operators in Subsection 2.1. We then present the `StoVoQ` compression scheme in Subsection 2.2. Finally, we compare `StoVoQ` to competing approaches, both theoretically and empirically on a small scale example with a high compression rate.

## 2.1 Unbiased gradient estimate to mitigate high compression rates

We here discuss an important property to mitigate high compression rates in FL settings. A *compression operator* Comp is a (random) mapping on $\mathbb{R}^d$. Consider the following assumption:

**A1 (Unbiased Compression with relatively bounded variance).** *A compression operator* Comp *is unbiased if for any $x \in \mathbb{R}^d$, $\mathbb{E}[\mathrm{Comp}(x)] = x$. It is said to have a $\omega$-bounded relative variance, for some $\omega > 0$, if it satisfies, for all $x \in \mathbb{R}^d$, $\mathbb{E}[\|\mathrm{Comp}(x) - x\|^2] \le \omega \|x\|^2$.*

83 The most classical compressors, especially `Q-SGD` and `Rand-`$H$ satisfy A 1 with different $\omega$, see
84 Subsection 2.3 and Table 1. On the other hand, some compression operators are biased, i.e.,
85 $\mathbb{E}[\text{Comp}(x)] \neq x$ for some $x \in \mathbb{R}$. Those operators are often deterministic, as is the case for
86 `Top-`$H$ compressor. The most classical assumption for biased operators, is the following contractive
87 property along the direction of descent [32, 5, 11]:

88 **A2** (**Biased Compression with contraction**). *For $\delta > 0$, a compression operator is said to be*
89 $1/(1 + \delta)$-*contractive if for any $x \in \mathbb{R}^d$, we have $\mathbb{E}[\|\text{Comp}(x) - x\|] \leq (1 - 1/(1 + \delta))\|x\|$.*

90 Constants $\omega$ and $\delta$ from these two assumptions are both positive, and become larger as the compression
91 rate increases. Alternative assumptions for the biased case have been introduced in [5].

92 **Impact of unbiasedness on the compression of a single vector.**[1] To understand the interaction be-
93 tween the number of workers $K$ and the compression error, a simple situation is the case in which the
94 workers use *independent and identically distributed compression operators* $(\text{Comp}_k)_{k=1}^K$ to compress
95 the *same vector* $x \in \mathbb{R}^d$. The central node aggregates $\{\text{Comp}_k(x)\}_{k=1}^K$ into $K^{-1} \sum_{k=1}^K \text{Comp}_k(x)$.
96 A bias-variance decomposition of the quadratic error gives:

$$\mathbb{E}[\|K^{-1} \textstyle\sum_{k=1}^K \text{Comp}_k(x) - x\|^2] = \|\mathbb{E}[\text{Comp}_1(x)] - x\|^2 + K^{-1}\|\mathbb{E}[\text{Comp}_1(x)] - x\|^2.$$

97 The variance of the aggregated vector is reduced by a factor $K^{-1}$ when averaging the messages
98 send by the $K$ workers, while the bias is independent of $K$. For example, if we use an unbiased
99 compressor satisfying A 1, we get

$$\mathbb{E}\left[K^{-1} \textstyle\sum_{k=1}^K \text{Comp}_k(x)\right] = x, \qquad \mathbb{E}\left[\left\|x - K^{-1} \textstyle\sum_{k=1}^K \text{Comp}_k(x)\right\|^2\right] \leq (\omega/K)\|x\|^2, \quad (2)$$

100 while for a deterministic biased compressor, we obtain that $K^{-1} \sum_{k=1}^K \text{Comp}_k(x) = \text{Comp}_1(x)$
101 has the same error as any of the individual compressed vector. We therefore pay particular attention
102 to obtaining an unbiased compressor in the following.

### 2.2 `StoVoQ` definitions and main properties.

104 The basic idea behind VQ is to quantize a vector
105 rather than each of its coordinates. A Vector
106 Quantizer is a mapping $\text{VQ}(\cdot, \mathscr{C}_M) : \mathbb{R}^d \to$
107 $\mathscr{C}_M$ which maps $x \in \mathbb{R}^d$ to an element of a
108 codebook $\mathscr{C}_M$, which is a finite subset of $\mathbb{R}^d$
109 with $M$ elements. The code of `StoVoQ` is pro-
110 vided in Algorithm 1, and its crucial steps are
111 described hereafter: we introduce the notion of
112 **(a)** Voronoi quantization scheme before describ-

---

**Algorithm 1:** `StoVoQ` with distribution $p$

**Input** : $x \in \mathbb{R}^d$, $p$, $M$, $P$, seed $s$
**Output :** Codeword index $\mathbf{i}_c$, value $\mathbf{i}_r$
1 Sample $\mathscr{C}_M \sim p$ with seed $s$ ;  /* generate codebook with distribution $p$ */
2 $c = \text{VQ}(x, \mathscr{C}_M^p)$;  /* perform Voronoi quant. */
3 $\mathbf{i}_c = $ index of $c$;  /* get index of codeword */
4 $r = r_M^p(\|x\|)$;  /* find radial bias in table */
5 $\mathbf{i}_r = \text{SQ}(r^{-1})$ ;  /* quantize $r$ on P bits */

---

113 ing more precisely **(b)** random codebooks, **(c)** whose distributions are invariant by unitary transforms.
114 Then, **(d)** a method to obtain an unbiased Voronoi scheme is presented and finally **(e)** its asymptotic
115 properties (as $M \to \infty$) are given.

116 **(a) Voronoi Quantization.** Voronoi quantization [23, 25], aims at selecting the closest codeword
117 from $\mathscr{C}_M$, i.e.:
$$\text{VQ}(x, \mathscr{C}_M) \triangleq \text{argmin}_{c \in \mathscr{C}_M} \|x - c\|. \qquad (3)$$
118 Unfortunately, for any given $\mathscr{C}_M$, the Voronoi quantizer is not *unbiased*: indeed it is deterministic
119 and $\text{VQ}(x, \mathscr{C}_M) \neq x$ if $x \notin \mathscr{C}_M$. A classical approach to construct a bias-free VQ is to use the
120 optimal "dual" VQ (or Delaunay quantization) [24], but this approach is numerically expensive (see
121 Subsection 2.3). To mitigate the bias, we rather use random codebooks.

122 **(b) Random Codebook.** A key ingredient of `StoVoQ` is the use of a random codebook within the
123 quantizer. We assume $\mathscr{C}_M = [C_1, \ldots, C_M]$ where *the codewords* $\{C_i\}_{i=1}^M$ are i.i.d. random vectors
124 distributed according to $p$, the codeword distribution pdf. We denote $\mathscr{C}_M \sim p$ and use boldface
125 to stress that $\mathscr{C}_M$ is random. When quantizing a sequence of vectors $\{x_t\}_{t=0}^\infty \subset \mathbb{R}^d$ we sample
126 for each $t \in \mathbb{N}$ a **new codebook** $\mathscr{C}_{M,t} \sim p$, compute $\text{VQ}(x, \mathscr{C}_{M,t})$ and transmit the index of the
127 corresponding codeword $i_{c,t} \in [M]$. The codebook $\mathscr{C}_{M,t}$ is **not transmitted**: the transmitter and the
128 receiver use the **same seeds** so that the same codebooks $\mathscr{C}_{M,t}$ can be reconstructed on both sides.

---
[1]The impact of unbiasedness for obtaining optimal convergence complexities in FL is discussed in Section 3.

129 **(c) Unitary invariant Codewords.** Denote by $\mathrm{U}(d) = \{U, U^*U = \mathrm{I}\}$ the set of unitary transforms
130 over $\mathbb{R}^d$. We assume in the sequel that the codeword distribution $p$ is unitary invariant, meaning that:
131 **A3.** *The distribution of the codewords $p$ is invariant under the unitary group, i.e. for all $U \in \mathrm{U}(d)$,*
132 *and any $x \in \mathbb{R}^d$, $p(Ux) = p(x)$.*

133 Examples of such distributions include isotropic Gaussian distributions ($p = \mathcal{N}(0, \sigma^2\,\mathrm{I}_d), \sigma^2 > 0$)
134 and the uniform distribution on the Sphere (which is specifically discussed in Appendix D.1). Under
135 A 3, there exists a non-negative function $p_{\mathrm{rad}}$ on $\mathbb{R}_+$ such that, for all $x \in \mathbb{R}^d$, $p(x) = p_{\mathrm{rad}}(\|x\|)$.

136 **(d) The quantization bias is radial.** Under A 3, we have the following crucial unitary invariance
137 property. For $A \subset \mathbb{R}^d$, and $U \in \mathrm{U}(d)$, we write $UA = \{Ux, x \in A\}$.

138 **Lemma 1.** *Assume A 3. For any nonnegative measurable function $f$, any $U \in \mathrm{U}(d)$ and $x \in \mathbb{R}^d$,*
139 $\mathbb{E}_{\mathscr{C}_M \sim p}[f(\mathrm{VQ}(Ux, \mathscr{C}_M))] = \mathbb{E}_{\mathscr{C}_M \sim p}[f(U\,\mathrm{VQ}(x, U\mathscr{C}_M))]$.

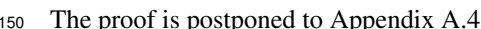

140 The proof is postponed to Appendix A.3. Tak-
141 ing $f(x) = x$, the previous result implies that
142 for any $x \in \mathbb{R}^d$ and $U \in \mathrm{U}(d)$, it holds that
143 $\mathbb{E}_{\mathscr{C}_M \sim p}[\mathrm{VQ}(Ux, \mathscr{C}_M)] = U\mathbb{E}_{\mathscr{C}_M \sim p}[\mathrm{VQ}(x, U\mathscr{C}_M)]$.
144 A direct consequence of the elementary Lemma 3 is
145 that the quantization error is radial:

146 **Theorem 1** (Quantization bias). *Assume A 3. Then,*
147 *for all $M \in \mathbb{N}$, there exists a function $r_M^p : \mathbb{R}_+ \mapsto$*
148 $\mathbb{R}_+$ *such that for all $x \in \mathbb{R}^d$, $\mathbb{E}_{\mathscr{C}_M \sim p}[\mathrm{VQ}(x, \mathscr{C}_M)] =$*
149 $r_M^p(\|x\|)x$.

150 The proof is postponed to Appendix A.4.

Figure 1: function $r_M^p$ for $d = 4$ (dashed) and $d = 16$ (solid), $p = \mathcal{N}(0, \mathrm{I}_d)$ and $M = 2^{10}$(orange), and $M = 2^{13}$(green).

151 In words, the expectation of the quantized vec-
152 tor $\mathrm{VQ}(x, \mathscr{C}_M)$ is *colinear* to the vector $x$, i.e.,
153 $\mathrm{VQ}(x, \mathscr{C}_M)$ is **directionally unbiased**. Moreover, this radial bias only depends on $\|x\|$, $M$ and
154 the distribution $p$. This function is intractable, but it is straightforward to pre-compute it using
155 Monte-Carlo method. We display $r_M^p$ for $p = \mathcal{N}(0, \mathrm{I}_d)$ in Figure 1. Consequently, we can remove
156 the bias of $\mathrm{VQ}(x, \mathscr{C}_M)$ by re-scaling the corresponding codeword by $1/r_M^p(\|x\|)$.

157 We now analyze the quantization distortion for a given $x \in \mathbb{R}^d$ vector. We need to strengthen the
158 assumption about the distribution of the codewords. Consider the following assumption

159 **A 4.** *(1) there exists $\epsilon > 0$ such that $\int r^{2+\epsilon} p_{\mathrm{rad}}(r)\mathrm{d}r < \infty$ (2) for some $\delta > 0$, $m_\delta =$*
160 $\inf_{r \leq \delta} p_{\mathrm{rad}}(r) > 0$, *and (3) $p_{\mathrm{rad}}$ is unimodal, i.e. the super level sets $\{r \in \mathbb{R}_+, p_{\mathrm{rad}}(r) \geq t\}$,*
161 *for $t \geq 0$ are convex subsets of $\mathbb{R}_+$.*

162 A 4 is obviously satisfied if we take $p = \mathcal{N}(0, \sigma^2\,\mathrm{I}_d)$ for any $\sigma^2 > 0$.

163 **Theorem 2.** *Assume A 3-A 4. Define $C_d = \pi^{-1}\Gamma(1 + 2/d)\Gamma(1 + d/2)^{2/d}$. Then, for every $x \in \mathbb{R}^d$,*

$$\lim_{M \to \infty} M^{2/d}\mathbb{E}_{\mathscr{C}_M \sim p}[\|\,\mathrm{VQ}(x, \mathscr{C}_M) - x\|^2] = C_d p_{\mathrm{rad}}^{-2/d}(\|x\|)\,.$$

164 The proof is postponed to Appendix C.1. Note that $C_d \approx_{d \to \infty} d/(2\pi\mathrm{e})$ hence $C_d$ grows only linearly
165 with the dimension $d$. We can now exploit this result to control the radial bias as a function of $\|x\|$.
166 Since $|r_M^p(\|x\|) - 1| \leq \|x\|^{-1}\{\mathbb{E}_{\mathscr{C}_M \sim p}[\|\,\mathrm{VQ}(x, \mathscr{C}_M) - x\|^2]\}^{1/2}$, Theorem 2 shows that

$$\limsup_{M \to \infty} M^{1/d}|r_M^p(\|x\|) - 1| \leq C_d^{1/2} p_{\mathrm{rad}}^{-1/d}(\|x\|)/\|x\|\,.$$

167 In other words, for any $x \in \mathbb{R}^d$, the radial bias $r_M^p(\|x\|)$ approaches 1 as $M \to \infty$ with a rate
168 $O(M^{-1/d})$. We use an a scalar quantizer SQ to transmit $1/r_M^p(\|x\|)$. Because the range of values
169 taken by $1/r_M^p(\|x\|)$ is limited, a small number of bits $P$ is sufficient (we typically use $P = 3$
170 bits). The total number of transmitted bits is $\log_2(M) + \log_2(P)$. We use a random unbiased scalar
171 quantizer (see e.g. [8, Eq. (2)]), a random mapping for $\mathbb{R} \to \mathcal{S}_P$ an ordered subset of $\mathbb{R}$ with $P$
172 elements. A scalar quantizer is said to be unbiased if $\mathbb{E}[\mathrm{SQ}(r)] = r$ for all $r \in \mathbb{R}$. Assuming that
173 SQ is independent of $\mathscr{C}_M$, we get for all $x \in \mathbb{R}^d$, $\mathbb{E}[\mathrm{SQ}(1/r_M^p(\|x\|))]\mathbb{E}_{\mathscr{C}_M \sim p}[\mathrm{VQ}(x, \mathscr{C}_M)] = x$. To
174 save space, we present the details of the scalar quantization (based on nonuniform random dither)
175 methods is presented in Appendix B.1.

176 **(e) Random vs. Optimal codebooks:** We finally motivate the choice of random codebooks and
177 describe how to choose the codevector distribution $p$. For a given pdf $q$ of the input the *(quadratic)*
178 *distortion* is defined as:

$$\mathrm{Dist}(q, \mathscr{C}_M) = \int_{\mathbb{R}^d} \|x - \mathrm{VQ}(x, \mathscr{C}_M)\|^2 \, q(x) \, \mathrm{d}x = \mathbb{E}_{X \sim q}[\|X - \mathrm{VQ}(X, \mathscr{C}_M)\|^2]. \qquad (4)$$

179 We stress that in this case the expectation is taken w.r.t. the input distribution $q$, the codebook
180 being deterministic in (4). A *Voronoi optimal codebook* $\mathscr{C}_M^{q,*}$ is a minimizer of the distortion over
181 the set of codebooks: $\mathrm{Dist}(q, \mathscr{C}_M^{q,*}) = \min_{|\mathscr{C}_M|=M} \mathrm{Dist}(q, \mathscr{C}_M)$. Zador's theorem [13] gives the
182 distortion of the Voronoi optimal codebook in the limit of $M \to \infty$; see Appendix C.1 for a precise
183 statement. Denote for $\beta \in \mathbb{R}_+$ and a function $f$ on $\mathbb{R}^d$, $\|f\|_\beta = (\int |f(x)|^\beta \mathrm{d}x)^{1/\beta}$. It is known that
184 if $\|q\|_{d/(d+2)} < \infty$, then as $M \to \infty$, $\mathrm{Dist}(q, \mathscr{C}_M) \approx M^{-2/d} J_d \|q\|_{d/(d+2)}$, and $J_d$ is a universal
185 constant $J_d$ satisfying $J_d \cong_{d \to \infty} d/2\pi\mathrm{e}$ (see Appendix C.2 for the exact constant).

186 Using Theorem 2, we can quantify the loss between random codebook distributed according to $p$ and
187 the Voronoi optimal codebook for a given input distribution $q$ when $M \to \infty$. Define

$$\mathrm{C}(q, p, d) = \int_{\mathbb{R}^d} p(x)^{-2/d} q(x) \mathrm{d}x. \qquad (5)$$

188 If $\|q\|_{d/(d+2)} < \infty$, using the Hölder inequality with negative exponents (see [15, p. 191] and
189 Appendix C.3),it holds that $\mathrm{C}(q, p, d) \geq \|q\|_{d/(d+2)}$.

190 **Theorem 3.** *Assume that $p$ satisfies A 3-A 4, $\|q\|_{d/(d+2)} < \infty$, $\int_{\mathbb{R}^d} \|x\|^{2+\delta} q(x) \mathrm{d}x < \infty$ for some*
191 $\delta > 0$, *and* $\mathrm{C}(q, p, d) < \infty$. *Then,*

$$\lim_{M \to \infty} \mathbb{E}_{\mathscr{C}_M \sim p}[\mathrm{Dist}(q, \mathscr{C}_M)] / \mathrm{Dist}(q, \mathscr{C}_M^{q,*}) = C_d J_d^{-1} \mathrm{C}(q, p, d) \|q\|_{d/(d+2)}^{-1}. \qquad (6)$$

192 *with $C_d$ defined in Theorem 2. Moreover, assume that input distribution $q$ satisfies A 3-A 4, and set the*
193 *codeword distribution $p_{q,d,*} = q^{d/(d+2)}(x) / \int q^{d/(d+2)}(x)\mathrm{d}x$. Then, $\mathrm{C}(q, p_{q,d,*}, d) = \|q\|_{d/(d+2)}$.*

194 The proof is postponed to Appendix C.2. In words, under general assumptions, the distortion
195 achieved by a random quantizer $\mathrm{VQ}(\cdot, \mathscr{C}_M)$, $\mathscr{C}_M \sim p$ is rate optimal (with rate $M^{-2/d}$). If
196 in addition $q$ is unitarily invariant and unimodal, then a random codebook distributed accord-
197 ing to $p_{q,d,*}$ reaches the optimal distortion bound, up to universal constants (depending only
198 on the dimension $d$). Moreover, as $d \to \infty$, then $C_d J_d^{-1} \cong_{d \to \infty} 1$ and the efficiency gap van-
199 ishes. As an illustration, assume that the input distribution is standard Gaussian $q = \mathcal{N}(0, \mathrm{I}_d)$
200 and set the codeword distribution to be $p_\alpha = \mathcal{N}(0, \alpha^2 \mathrm{I}_d)$ where $\alpha^2 \in \mathbb{R}_+^*$. If $\alpha^2 d > 2$, then
201 $\mathrm{C}(\mathcal{N}(0, \mathrm{I}_d), \mathcal{N}(0, \alpha^2 \mathrm{I}_d), d) = 2\pi\alpha^2 \{\alpha^2 d / (\alpha^2 d - 2)\}^{d/2}$ and $\|\mathcal{N}(0, \mathrm{I}_d)\|^{(2+d)/2} = (2\pi)(1 +$
202 $2/d)^{1+2/d}$. The function $\alpha \to \mathrm{C}(\mathcal{N}(0, \mathrm{I}_d), \mathcal{N}(0, \alpha^2 \mathrm{I}_d), d)$ has a unique minimum at $\alpha_d^2 = 1 + 2/d$
203 for which $\mathrm{C}(\mathcal{N}(0, \mathrm{I}_d), \mathcal{N}(0, \alpha_d^2 \mathrm{I}_d), d) = \|\mathcal{N}(0, \mathrm{I}_d)\|^{(2+d)/2}$ showing that a random codebook sam-
204 pled from $\mathcal{N}(0, \alpha_d^2 \mathrm{I}_d)$ is optimal. It is interesting to note that the variance of the codeword distribution
205 should be $(1 + 2/d)$ larger than the variance of the input distribution $\mathcal{N}(0, \mathrm{I}_d)$.

## 2.3 Related works

207 We compare `StoVoQ` with competing (random) compressors; additional details are given App. A.1.

208 **QSGD.** Alistarh et al. [2] compresses each coordinate of the scaled vector $x/\|x\|$ on $s + 1$ codewords.
209 QSGD is a scalar quantizer which requires $\mathcal{O}(\sqrt{d} \log_2(d))$ bits in its highest compression setting
210 ($s = 1$, only two possible levels for each coordinate). The vector norm is transmitted with full
211 precision $\|x\|$ (16 or 32 bits). This is in general substantially higher than the number of bits used by
212 VQ methods. In deep learning problems, it reduces the communication cost by a factor of 4 to 7 [2,
213 Sec. 5].

214 **Top-H/Rand H.** Achieving higher compression rates is possible through *sparsification* operators, that
215 only transmit a few coordinates. The most popular schemes are `Top-`$H$ and `Rand-`$H$ compressors,
216 that respectively map the vector to either its $H$ largest coordinates, or a random subset of cardinality
217 $H$, rescaled by $d/H$ to ensure unbiasedness. `Top-`$H$ is a biased operator, and the performance of
218 `Rand-`$H$ are poor on deep learning tasks [5, Figures 4 and 5].

Table 1: Per iteration communication complexity of most frequently used algorithms in dimension $d$. Constants $H$ and $M$ respectively correspond to a number of coordinates to be transmitted and a number of codewords, they are chosen by the user.

| | Uncomp. | Scalar Quantization | | | | Vector Quantization | | | | |
|---|---|---|---|---|---|---|---|---|---|---|
| | SGD | Sign | QSGD$_{s\geq 1}$ | Top-$H$ | Rand-$H$ | Polytope [10] | HSQ-span [8] | HSQ-greed [8] | StoVoQ | DoStoVoQ |
| #bits | $32d$ | $d$ | $32 + s\sqrt{d}\log_2(d)$ | $32H$ | $32H$ | $\log_2(2d)$ | $\log_2(M)$ | $\log_2(M)$ | $\log_2(M)$ | $\log_2(M)$ |
| Unbiased | - | | ✓ | | ✓ | ✓ | ✓ | | ✓ | ✓ (Th.4) |
| A.1 ($\omega + 1$) | - | - | $\sqrt{d}/s$ | - | $d/H$ | $d$ | $d$ | - | | $O(M^{-2/d})$ (Th.4) |
| A.2 ($\delta + 1$) | - | - | - | $d/H$ | - | - | | $M/\sigma_{\min}(C)$ | - | - |

**HyperSphere Quantization (HSQ).** HSQ was introduced by Dai et al. [8]. Two versions are considered: (1) a - greedy- Voronoi VQ referred to as HSQ-greed in Table 1, which is biased, and for which the theoretical guarantee provided in the paper (in their Lemma 3 and Theorem 3, which corresponds to a variant of A 2 and the subsequent convergence rate) *worsens* as $M$ increases, making it mostly vacuous; (2) an unbiased version VQ (HSQ-span), which uses a minimum-norm decomposition of $x \in \text{Span}(\mathscr{C}_M)$ the linear subspace generated by the codewords - this version suffers from a large variance (see Table 2) and potentially an ill-conditioning. Moreover, the performance of HSQ-span does not improve with $M$.

StoVoQ builds on HSQ-greed, that achieves high compression factors (up to 60-100 to obtain close to SOTA performance on CIFAR10), while preserving a good flexibility w.r.t. the compression level. StoVoQ approach allows to remove its inherent bias and provide a much stronger convergence analysis: **our approach is the first vector quantization scheme to provably benefit from an increasing number of elements in the codebook** $M$ (and obviously benefits from the number of workers $K$, as it is unbiased).

**Dual Quantization and Cross-polytope.** An approach to constructing unbiased VQ is to use the dual VQ, also referred to as Delaunay Quantization (DQ); see [24]. DQ is unbiased for any $x \in \text{ConvHull}(\mathscr{C}_M)$, the convex hull of $\mathscr{C}_M$. DQ requires to compute the barycentric coordinates for $x \in \text{ConvHull}(\mathscr{C}_M)$, that is to solve $(\lambda_1^x, \ldots, \lambda_M^x) = \text{argmin}_{\lambda_1, \ldots, \lambda_M} \|x - \sum_{i=1}^M \lambda_i c_i\|^2$, under the constraints $\lambda_i \geq 0, \sum_{i=1}^M \lambda_i = 1$. The quantizer is obtained by drawing a codeword $c_i$ with probability $[\lambda_1^x, \ldots, \lambda_M^x]$. Computing the barycentric coordinates is in general very demanding unless $\mathscr{C}_M$ has a very simple structure (see Appendix B for details). The Cross-Polytope method Gandikota et al. [10] is a simple instance of DQ, with a codebook $\mathscr{C}_{2d}^{\text{CP}}$ composed of the $2d$ canonical vectors $\{\pm\sqrt{d}e_i = \pm(0, \ldots, 0, \sqrt{d}, 0 \ldots 0), i \in [d]\}$, that relies on the inclusion $\text{B}_2(0; 1) \subset \text{B}_1(0; \sqrt{d}) = \text{ConvHull}(\mathscr{C}_{2d}^{\text{CP}})$. The barycentric decomposition can then easily be computed. Unfortunately, this method suffers from a large variance, as the quantization error $\|\text{VQ}^{\text{CP}}(x, \mathscr{C}_M) - x\|$ of *any* $x$ is *lower bounded* by $\sqrt{d} - 1$, which means the error has the same quadratic error than the Rand-1 compressor.

Table 1 summarizes the number of bits required to exchange the compressed value of a vector $x \in \mathbb{R}^d$ for the compression methods considered in this Section, as well as the assumptions they satisfy.

**Numerical comparisons:** In Table 2, we compare the distortions achieved by the compression methods given in Table 1 for a communication budget of 16 bits for $d = 16$ and assuming that the input distribution is $q = \mathcal{N}(0, \text{I}_d)$. The compression factor is 32 (assuming 32 bits floating point per coordinate). Such a compression rate is out of reach for QSGD, that requires, even for $s = 1$ at least $\sqrt{d}\log(d) + R$ bits, where $R$ is the number of bits to encode the norm (32 in [2]). For QSGD we have quantized the norm (using an uniform quantizer) on 3 bits and obtained an averaged distortion of 36.10 (for $K = 1$) and 1.82 for ($K = 20$) - the total number of bits is 19-. We use $H = 2$ for Top-$H$ and Rand-$H$ and use a scalar quantizer with 8 bits. For HSQ, we use 6 bits for the norm, using the unbiased uniform quantizer given in [8] and a Voronoi optimal codebook for the uniform distribution on the unit-sphere with $M = 2^{10}$ codewords. For StoVoQ we use a random codebook with $M = 2^{13}$ codewords; the codewords are sampled from a $\mathcal{N}(0, (1 + 2/d)\,\text{I}_d)$, and 3 bits are allocated for the scalar quantization of $1/r_M^p$ (the inverse of the radial bias). Finally, we average the result of 2 independent compressions for Polytope (following the replication technique described in [10]). We use $n = 10^4$ vectors, and report in Table 2 the distortion and sample variance. For StoVoQ with $K = 20$, the codebooks of the different workers are independent.

Table 2: Distortion for Gaussian inputs, for a fixed budget of 16 bits with $d = 16$.

| Method | Sign [4] | Top-2 | Rand-2 | Polytope [10] | HSQ-span [8] | HSQ-greed [8] | StoVoQ |
|---|---|---|---|---|---|---|---|
| # Bits (obj =16) | 16 | $2 \times 8$ | $2 \times 8$ | $\log_2(2 \times 16) \times 2 + 6$ | $\log_2(2^{10}) + 6$ | $\log_2(2^{10}) + 6$ | $\log_2(2^{13}) + 3$ |
| Unbiased | | | ✓ | ✓ | ✓ | | ✓ |
| $K = 1$ | 6.21 (0.02) | 8.40 (0.04) | 102.8 (0.9) | 113.9 (0.6) | 146.9 (0.6) | 9.03 (0.04) | 6.97 (0.02) : |
| $K = 20$ | 6.26 (0.02) | 8.76 (0.04) | 5.40 (0.04) | 5.98 (0.03) | 7.58 (0.04) | 9.10 (0.04) | 0.838 (0.005) |

## 3   `DoStoVoQ` **algorithm**

We illustrate how the `StoVoQ` compression scheme can be implemented in FL. To avoid cumbersome technical details, we focus here on the `Federated-SGD` algorithm. At iteration $t + 1$, each worker computes a stochastic gradient $g_{k,t+1}$ of the loss $f_k$ at the current model $\theta_t$, compresses it into $\hat{g}_{k,t+1} = \text{Comp}(g_{k,t+1})$ and send it to the central server, that performs the update step $\theta_t = \theta_{t-1} - \gamma_t/K \sum_{k=1}^{K} \hat{g}_{k,t}$. The code of the resulting algorithm, `DoStoVoQ-SGD`, is given in Algorithm 2. At iteration $t + 1$, the crucial steps are:

1. Worker $k \in [K]$ computes the norm $\|g_{k,t+1}\|$ of the $D \times 1$ gradient $g_{k,t+1}$ and then splits the scaled gradient $g_{k,t+1} \times \sqrt{D}/\|g_{k,t+1}\|$ into $L$-buckets of size $d$: $g_{k,t+1} \times \sqrt{D}/\|g_{k,t+1}\| = [b_{k,t+1}^1, \ldots, b_{k,t+1}^L]$. The norm $\|g_{k,t+1}\|$ is transmitted to the central node using a high-resolution scalar quantizer (or without quantization).

2. Each worker quantizes the buckets $\{b_{k,t+1}^1, \ldots, b_{k,t+1}^L\}$ using `StoVoQ`. **Independent** codebooks $\{\mathscr{C}_{M,k,t+1}\}_{k\in[K]}$ are used to ensure that the quantizers remain conditionally independent (see below for a precise statement). The double stochasticity (each worker uses random codebooks, which are independent between workers and across iterations) motivates the name `DoStoVoQ`. At iteration $t$, the same codebook is used for all buckets of worker $k$. Formally, for $\ell \in [L]$ we apply (in parallel) `StoVoQ`$(b_{k,t+1}^\ell, p, M, P, s_{k,t+1})$, with a sequence of different seeds $(s_{k,t+1})_{k\in[K],t\geq0}$. This sequence is shared between the workers and the central node at initialization.

3. The central node computes $(\hat{g}_{k,t+1})_{k\in K}$ from all messages received, performs the update on $(\theta_t)_{t\geq}$, and broadcasts $\theta_{t+1}$ to the workers.

These steps would similarly allow to incorporate `StoVoQ` within any of the advanced FL algorithms, and Theorem 4 is the crucial assumption to derive the convergence rates, as described in Section 2. Natural extensions to `DoStoVoQ-Fed-Avg`, `DoStoVoQ-DIANA` and `DoStoVoQ-VR-DIANA` are provided in Appendix D.2.

**Bias and variance of the compressed gradient with $K$ workers.** Consider the two filtrations $(\mathcal{F}_t)_{t\geq0}$ and $(\mathcal{G}_t)_{t\geq0}$ defined recursively as follows $\mathcal{F}_0 = \sigma(\emptyset)$ and for $t \geq 0$, $\mathcal{G}_{t+1} = \mathcal{F}_t \vee \sigma(\{g_{k,t+1}, k \in [K]\})$ and $\mathcal{F}_{t+1} = \mathcal{G}_{t+1} \vee \sigma(\{\hat{g}_{k,t+1}, k \in [K]\})$. With these notations, for any $t \geq 0$, $\theta_t$ is $\mathcal{F}_t$-measurable.

**Theorem 4.** *At any iteration $t + 1$ in DoStoVoQ, the $K$ compressed stochastic gradients $(\hat{g}_{k,t+1})_{k\in[K]}$ are (i) independent conditionally to $\mathcal{G}_{t+1}$ (ii) conditionally unbiased, i.e., for all $k \in [K]$, we have $\mathbb{E}\left[\hat{g}_{k,t+1} \mid \mathcal{G}_{t+1}\right] = g_{k,t+1}$, (iii) satisfy the relatively bounded error condition of A 1, i.e. there exists a constant $\omega_M$ such that, for all $k \in [K]$: $\mathbb{E}\left[\|\hat{g}_{k,t+1} - g_{k,t+1}\|^2 \mid \mathcal{G}_{t+1}\right] \leq \omega_M \|g_{k,t+1}\|^2$.*

---

**Algorithm 2:** `DoStoVoQ-SGD` over $T$ iterations

**Input** : $T$ nb of steps, $(\gamma_t)_{t\geq0}$ LR, $\theta_0, p, M, P$ ;
**Output** : $(\theta_t)_{t\geq0}$
1 **for** $t = 1, \ldots, T$ **do**
2    $w_0$ sends $\theta_{t-1}$ and different seeds $s_{k,t}$ to each $w_k$;
3    **for** $k = 1, \ldots, K$ **do**
4      Compute local gradient $g_{k,t}$ at $\theta_{t-1}$;
5      Split $g_{k,t} \times \sqrt{D}/\|g_{k,t}\|$ on $[b_{k,t}^1, \ldots, b_{k,t}^L]$ ;
6      **for** $\ell = 1, \ldots, L$ *(in parallel)* **do**
7        $(\mathbf{i}_c^{t,k,\ell}, \mathbf{i}_r^{t,k,\ell}) = $ `StoVoQ`$(b_{k,t}^\ell, p, M, P, s_{k,t})$
8      **end**
9      Send $(\|g_{k,t}\|, (\mathbf{i}_c^{t,k,\ell}, \mathbf{i}_r^{t,k,\ell})_{\ell\in[L]})$ to $w_0$ ;
10    **end**
11    Reconstruct $(\hat{g}_{k,t})_{k\in K}$ ;
12    Update: $\theta_t = \theta_{t-1} - \gamma_t \frac{1}{K} \sum_{k=1}^{K} \hat{g}_{k,t}$ ;
13 **end**

---

*Moreover, $\omega_M$ decreases with the number of codewords $M$ and the $P$, as $\omega_M = O(M^{-2/d}) + O(2^{-P})$ [the dependence on $p$, $d$, and $D$ is made explicit in the proof].*

308 The first statement stems from the fact that each bucket is quantized using `StoVoQ` which is unbiased.
309 The second statement is more challenging; proof is postponed to Appendix A.6. We stress that this
310 result differs from Theorem 2, which corresponds to the distortion of a source with distribution $q$.

311 **Convergence results.** Theorem 4 proves that our compression method satisfies the assumptions
312 needed to obtain fast convergence rate, for `DoStoVoQ-SGD`, and for its variants `DoStoVoQ-(VR)`-
313 DIANA. Consider a Smooth and Strongly Convex (SSC) function $F = \sum_{k=1}^{K} f_k$, with condition
314 number $\kappa > 1$. We measure the complexity of the algorithm by the number of iterations $t$ required
315 to obtain a model $\theta_t$ such that $\mathbb{E}[F(\theta_t)] - \min_{\mathbb{R}^D} F \leq \epsilon$. The result of VR-DIANA [16], which
316 provides a complexity of $O_{\kappa \to \infty} \left( \kappa \left(1 + \omega_M/K\right) \log(\epsilon^{-1}) \right)$ [16, Corollary 2], applies to `DoStoVoQ`-
317 VR-DIANA.

318 Convergence rates for `DoStoVoQ-DIANA` (without VR), and on non-convex optimization problems
319 can be obtained from Horváth et al. [16, Corollary 1,3,4]. As in the strongly-convex case, complexities
320 increase by a factor depending on $(1 + \omega_M/K)$ w.r.t. uncompressed algorithm. Intuitively, *the impact*
321 *on the optimization complexity of a high compression is mitigated by the number of workers*, which
322 supports the use of independent and unbiased compressors when the number of workers is large and
323 high compression factors are required.

324 Indeed, these complexities can be compared to: (1) the one of *uncompressed* variance reduced
325 distributed methods [9] that achieve a complexity of $O_{\kappa \to \infty} \left( \kappa \log(\epsilon^{-1}) \right)$ (in the SSC case); (2) the
326 complexity for biased compression operators satisfying A 2, Beznosikov et al. [5, Theorem 13] that
327 obtain $O_{\kappa \to \infty}(\kappa(1 + \delta) \log(\epsilon^{-1}))$ for compressed GD (independently of the number of workers);
328 (3) the complexities of compressed SGD methods with *error feedback* in [11][2], that also have no
329 dependency on the number of workers. **Overall, the unbiased character is crucial to mitigate the**
330 **variance increase resulting from high compression rates.**

# 4 Numerical experiments

## 4.1 Least Squares Regression (LSR)

We consider a least-squares problem with $n =$
$2^{14}$ samples, a bucket size $d = 16$, $D = 2^9$, and
$K = 32$ workers; each worker has access to a
subset $m = 2^{11}$ samples (picked with replace-
ment) to introduce a dependency in the data used
by the workers. For $i \in [n]$, we assume $X_i \sim$
$\mathcal{N}(0, \mathrm{I}_D)$ and $Y_i \sim \mathcal{N}(X_i^\top \omega_*, 1)$ where $\omega_* \in$
$\mathbb{R}^D$. We solve $\inf_{\omega \in \mathbb{R}^D} \sum_{i=1}^{n} \|Y_i - X_i^\top \omega\|^2$ via
a gradient descent with step size $1/\alpha L$ where
$\alpha$ is fine-tuned for each quantization method
and $L \approx 2n$ is the smoothness constant. We
use `DoStoVoQ` with $M = 2^{13}$ codewords sam-
pled from $\mathcal{N}(0, (1 + 2/d)\,\mathrm{I}_d)$ for `DoStoVoQ` and
$M = 2^{10}$ on the unit Sphere for `HSQ` s.t. the
number of bits transmitted at each round by the
worker is set to 16 (see Table 2). Figure 2 reports

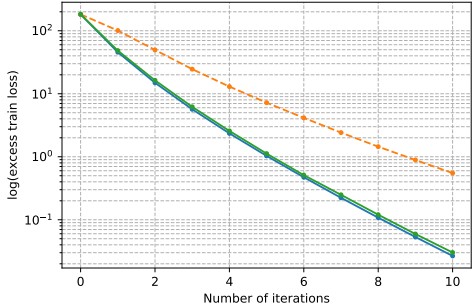

Figure 2: Comparison between GD (blue),
`HSQ-greed` (orange) and `DoStoVoQ` (green), on
a LSR problem in dimension $D = 2^9$.

the excess-log of the train loss over $T = 10$ iterations, for a standard GD. `DoStoVoQ` outperforms
`HSQ-greed`: indeed the linear convergence rate of distributed GD is faster for an unbiased compressor
than for the biased approach.

## 4.2 Applications to Deep Neural Networks training

**Setting.** We now describe our experimental framework for training two standard models of Deep
Neural Networks: a VGG-16 [31] and a ResNet-18 [14]. We follow the standard procedure of training
those models both on CIFAR-10 and ImageNet; the hyper-parameters are fine-tuned to optimize the
accuracy *without quantization*. We do not compress the affine constant part of the affine convolutional

---

[2]authors provide complexities for 10 algorithms in Table 1, with Error Feedback and under A 2.

Table 3: Average accuracy over 5 experiments, after 100 epochs on CIFAR-10.

| Algorithm | SGD | QSGD 2 bits | QSGD 4 bits | QSGD 8 bits | HSQ $d = 16$ | HSQ $d = 8$ | Dos. $d = 16$ | Dos. $d = 8$ |
|---|---|---|---|---|---|---|---|---|
| Raw bits per bucket | $32d$ | | $\sqrt{d}\log(d)$ | | | $\log(d)$ | | |
| Effective Compression factor | 1 | $\sim 13$ | $\sim 8$ | $\sim 4$ | 34 | 17 | 38 | 20 |
| $K = 1$ worker | 91.9 | 91.7 | 92.1 | 91.9 | 92.0 | 92.0 | 92.0 | 92.1 |
| $K = 8$ worker | 92.0 | 91.8 | 91.8 | 92.0 | 91.8 | 92.0 | 91.8 | 92.1 |

Table 4: Distortion for on a subset $\mathcal{G}$ of the gradients of a layer of CIFAR-10, for a fixed budget of 16 bits with $d = 16$.

| Method | Top-2 | Rand-2 | Polytope [10] | HSQ-span [8] | HSQ-greed [8] | DoStoVoQ |
|---|---|---|---|---|---|---|
| # Bits (obj =16) | $2 \times 8$ | $2 \times 8$ | $\log_2(2 \times 16) \times 2 + 6$ | $\log_2(2^{10}) + 6$ | $\log_2(2^{10}) + 6$ | $\log_2(2^{13}) + 3$ |
| Unbiased | | ✓ | ✓ | ✓ | | ✓ |
| $K = 1$ | 0.0022 | 0.025 | 0.028 | 0.034 | 0.0021 | 0.0026 |

layers and batch normalization layers. We apply independent `DoStoVoQ` on batches of 32 buckets of size $d = 16$ (i.e. we transmit a high-resolution norm for $D = 32 \cdot 16 = 512$ coefficients).

**CIFAR-10.** We use the implementation of HSQ [8]: the batch size is 256 for CIFAR-10, the total number of epochs is 100, the initial learning rate is 0.1, which is divided by 10 and 50 at epochs 51 and 71. We report the accuracy of `DoStoVoQ`, `QSGD`, and `HSQ-greed` in table 4. By design, the compression factor of `Q-SGD` for $d = 16$ is 13, which is significantly less than HSQ or `DoStoVoQ`. Both `HSQ` and `DoStoVoQ` perform similarly and the accuracy gap between the two methods are under the sample variance (computed over 5 seed and about 0.2). In Table 4 we report the distortion of a random subset of gradients $\mathcal{G} = \{g_t, t \in [|\mathcal{G}|]\}$ (with $|\mathcal{G}| = 10^2$, $d = 16$, $D = 2^5 \times d$) obtained from a given layer of a VGG on CIFAR-10, i.e.: $|\mathcal{G}|^{-1} \sum_{g_t \in \mathcal{G}} \left\| K^{-1} \sum_{k=1}^{K} (g_{k,t} - \hat{g}_{k,t}) \right\|^2$, where $(\hat{g}_{k,t})_{k \in [K]}$ correspond to $k$ independent workers compressing their own gradient $g_{k,t}$. The choice of the layer does not affect significantly the results. Even with the actual gradient distribution, `DoStoVoQ` outperforms for a given compression factor each unbiased method. This is on pair with the observation that the gradients of a Deep Neural Network are approximately Gaussian distributed [3, 36, 4]. Additional experiments can be found in the Appendix.

**ImageNet.** For ImageNet, we use different bucket sizes, the standard batch size of 256, and only $K = 1$ worker for energy savings (recall Imagenet training last about 1 day for a single worker on academic hardware). An initial learning rate of 0.1 is divided by 10 at epoch 30 and 60, while the model is trained for 90 epochs. A ResNet here obtains 69.9%, and with a compression factor of 8, the performance drops by 2.5%. Using $d = 16$, we reach a compression factor of 38, while the Top-1 accuracy drops by only 4.8%: this is a substantially higher compression rate than the concurrent work `QSGD` on the ImageNet dataset.

**Computational impact.** In the case of deep Neural Networks, our training procedure requires neither a substantial modifications of standard pipelines, nor a modification of the hyper-parameters which allows to save computational resources. `Green Algorithm` ([20]) shows that this work generated around 15kg of CO2, and require 400 kWh. A typical experiment lasted few hours on CIFAR-10 and about 3 days on ImageNet, which is in the standard range for this type of prototypical codes. This work could have future impact on FL, to reduce their electrical consumption.

**Broader impact.** Federated learning enables multiple actors to build a common model without data sharing, hence respecting privacy. However classic FL methods consume an important amount of energy in transmitting information. Our method `DoStoVoQ` can be adapted to any FL framework while enabling important bandwidth savings. These savings highly counterbalance the computational impact of our experiments.

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
