# OpenReview forum: "$\texttt{DoStoVoQ}$: Doubly Stochastic Voronoi Vector Quantization SGD for Federated Learning"
_NeurIPS.cc/2021/Conference — NeurIPS 2021 Submitted_

### Official Review · Reviewer_jQuP · 2021-07-16

**Rating:** 5
**Confidence:** 4

**Summary:**

This paper presents DoStoVoQ, a vector quantization method based on Voronoi quantization. It is claimed that the compression method is designed specifically for federated learning applications. Both theoretical analysis and empirical results are provided to show the effectiveness of DoStoVoQ.

**Limitations And Societal Impact:**

My major suggestions are summarized in the “Cons” of “Main review”. Please take a close look at it. Overall, the proposed method can be improved to attain better results on real-world large-scale ML tasks.

**Main Review:**

Pros:
1. The paper is well-written and the idea is easy to follow. Improving the communication efficiency of federated learning is a promising research direction.
2. The scales of the experiments are large enough to provide valid evidence on the performance of DoStoVoQ.

Cons:
The major concern is that by looking at the results, I am not sufficiently convinced that DoStoVoQ works well in practice, e.g., it leads to 2.5% accuracy loss on ImageNet (line 376), which is not acceptable in practice. In my view, the latest communication efficiency method achieves almost the same final accuracy and does not require compressing the gradient [1] (though DoStoVoQ has the potential to achieve higher compression rates). Maybe one way to improve DoStoVoQ is to adopt the error feedback scheme, but more experiments are required to be done.

Other concerns are elaborated below:
1. It is claimed that DoStoVoQ is for FL. But both of the formulation and experiment settings are about normal distributed SGD. How to make DoStoVoQ compatible with FedAvg/FedProx [2-3]? How does DoStoVoQ perform under data heterogeneity?

2. Vector quantization is not new in communication efficient distributed training. How does DoStoVoQ compare to [4]?

3. In some sense, DoStoVoQ is similar to the low-rank based compression methods, e.g., PowerSGD, GradiVeQ and Atomo [5-7]. How does DoStoVoQ compare to them?

4. As mentioned above, how to make DoStoVoQ compatible with error feedback? Intuitively, it should be easy to achieve.

5. In the experiments, only compression rates are reported. Please also report the end-to-end wall clock time.

Minor comments:
Missing references [6-7].

[1] https://arxiv.org/abs/2103.03936

[2] https://arxiv.org/pdf/1602.05629.pdf

[3] https://arxiv.org/abs/1812.06127

[4] https://arxiv.org/abs/1911.07971

[5] https://arxiv.org/abs/1905.13727

[6] https://arxiv.org/pdf/1811.03617.pdf

[7] https://arxiv.org/abs/1806.04090


=========== Post rebuttal ==============
After carefully reading the authors' response, part of my concerns is addressed. Thus, I increase my overall evaluation score from 4 to 5. However, I still have the concern on the real speedup that the proposed method can achieve. Please refer to my comments below.
====================================

**Time Spent Reviewing:**

2.5

---

> ### Author Response · Authors · 2021-08-10
> **Response to Reviewer jQuP**
>
> We thank the reviewer for his/her time, careful reading, and insightful comments.  The paper will be revised accordingly. We are glad that the reviewer found the paper well written and easy to follow, and that he/she underlined the importance of this research direction. We hereafter provide a detailed answer to each question raised, and apologize for the long rebuttal.
>
>
> **Adaptation to FedAvg, FedProx**
> Our compression operator can be coupled with any (compressed) learning algorithm, the same way that StoVoQ is used into DoStoVoQ-SGD (that corresponds to a distributed compressed SGD). We instantiated two more examples in the Appendix, DIANA and VR-DIANA.
> We chose to focus on these three algorithms in the paper *because they were originally proposed with compression* [1,2,3], thus the convergence rates in those papers directly apply to our algorithm.
> Yet, extension to FedAvg, Fedprox, SCAFFOLD, etc., that were introduced without compression, is straightforward, and guarantees would also be applicable (see e.g., [5] for some rates of FedAvg and compression).
>
> We will highlight the fact that the method can be applied within any FL-algorithm.
>
>
>
> **Performance under heterogeneity.** We will add experiments with heterogeneity to the paper. As algorithms with compression and without "DIANA-memory" are known not to perform well (independently of which compressor is used), thus we performed a complementary experiment this week, running DoStoVoQ-DIANA  (Alg. 5 in Appendix).
>
> CIFAR, VGG16 | Compression | Accuracy |
> --- | ---:| ---:|
> [new]  DoStoVoQ-DIANA, d=8, $M=2^{12}$ | 20x | 90.75\%|
> [new]  SGD (baseline) | no compression  | 90.73\% ||
>
> This shows the robustness of our approach to heterogeneity. To obtain heterogeneity, for each worker, 50\% of the data was selected from a unique class (different for each worker), and 50\% uniformly among all classes.
>
> **Compatibility with EF.** Adding EF is also straightforward (that is, the modification is independent of the compression technique). We chose not to add this mechanism to preserve our focus on the compression technique itself, and because the importance of EF is mostly supported on biased contractive operators [6,7,8].
>
> In practice, EF is known to often improve convergence. We performed a supplementary experiment on CIFAR, with the same tuning, which allowed to improve the performance by 0.6\% to 92.7\%!
>
> CIFAR, VGG16 | Compression | Accuracy |
> --- | ---:| ---:|
> [new] DoStoVoQ-SGD-**EF**, d=8, $M=2^{12}$ | 20x | 92.7\%|
> DoStoVoQ-SGD, d=8, $M=2^{12}$ | 20x | 92.1\% ||
>
>
> We will add these results to the paper. However, we also want to underline that an important message of our paper is that focusing on simpler metrics (e.g., distortion) can possibly bring better insights to the community than "aggregated" metrics (performance of an advanced algorithm, combining memory and EF, after 100 epochs on a large dataset, with a specific batch size, learning rate decay, momentum parameter, etc.), that may not reflect the actual quality of the compression technique, or highly depend on the computational power used for tuning.
>
>
> **Performance on ImageNet.**
> Similarly, on ImageNet, our goal was **not** to achieve the baseline accuracy, but to show that we could achieve **much higher compression rates** while getting a "reasonable performance". Especially, we stress that we performed **no parameter tuning** for the DoStoVoQ run, and used exactly the same parameters than the ones **optimized for SGD**. Indeed, theoretical perspectives suggest that the optimal learning rate changes with compression. However, we have aimed at being as much conservative as possible, and tried to not optimize it for obtaining a fair comparison with competitive works. Improving upon those results would require to perform a more careful analysis of our gradient estimate with the "simple metric" that we have introduced. Yet, we believe this is beyond the scope of this paper and would require a specific analysis on its own.
>
>
> Similarly, our experiments are performed without adding Error-Feedback (that resulted in an improvement on CIFAR10) and is widely used in practice for methods achieving the best performance [9]. We chose not to focus on this technique which is orthogonal to our contributions and only theoretically supported for biased compression operators.
>
> Finally, we would like to highlight that many papers (e.g, Power-SGD, Atomo, GradiVeq, ...)  proposing new compression techniques did not perform experiments on ImageNet.
>
>
> **Comparison to [4] (vqSGD paper).** Several methods were proposed in this article: we already carefully compared to the Cross-Polytope scheme, which is the one on which most of the focus is placed and on which all experiments are performed in [4]. Especially, we reported distortion results for this scheme in Tables 1, 2 and 5 to 8 in the appendix, and provided detailed explanation on the reason why this scheme has a high variance (Fig. 4 in the appendix).
>
> To be thorough, we performed the same experiment on CIFAR10 than with our compression technique. To obtain the same compression rate, we used the replication technique described in [4, Sec. 7].
>
> CIFAR, VGG16 | Compression | Accuracy |
> --- | ---:| ---:|
> [new] Cross-Polytope, d = 8,  | 16x | 90.9\% |
> DoStoVoQ-SGD, d=8, $M=2^{12}$ | 20x | 92.1\% |
> SGD (baseline)  | no compression  | 91.9\% ||
>
>
> More generally, all methods in [4] are based on (i) working on a bounded domain $B(0,R)$, (ii) using a fixed (possibly generated from a distribution) set of points $C$, whose convex hull contains contains $B(0,1)$, and computing a decomposition of any vector of the unit ball over that set to obtain an unbiased random decomposition. Computing such a decomposition is extremely expensive ($O(\text{card}(C)^3)$, Remark 3 in [4]). One example is a Gaussian point set, but it requires a number of points that grows exponentially with the dimension, which makes the computation of the decomposition nearly intractable. The paper does not provide  experiments on that setting and our approach is completely orthogonal.
>
>
> **Link to PowerSGD, GradiVeQ.**  While the goal of these papers is similar (proposing a novel compression approach), there are major differences. First, these two papers come without any theoretical guarantee. Secondly, they result in a biased compression scheme, and thus do not fully benefit from an increase in the number of workers. Finally, they are supported by a "model" of the gradients to compress (low rank for PowerSGD, highly correlated for GradiVeq). On the other hand, our method does not make any such assumption on the gradients distribution. We will expand the discussion on those methods.
>
> **Link to Atomo.** Atomo is mostly related to some of the methods in [4], where the goal is to compute an unbiased decomposition of a vector on a set of points. As underlined in our paper, this corresponds to a Delaunay decomposition. This requires to solve a meta-optimization problem at each step, which can result in substantial computational overhead. Finally, the best performance reported on CIFAR10 and a ResNet-18 is 80\% on test accuracy (Fig 3a). We will add the reference and comments.
>
> CIFAR-10, **Resnet 18** | Compression | Accuracy |
> --- | ---:| ---:|
> [Atomo, Fig 3a] Atomo, | $6\times$ $\ ^1$  | 80\%|
> [New] DoStoVoQ-SGD, d=8, $M=2^{12}$ | $20\times$ | 94.35\%||
>
>
> CIFAR-10  | Compression | Accuracy |
> --- | ---:| ---:|
> [Atomo, Fig 3c] Atomo, **VGG11** | $6\times$ $\ ^1$   | 78\%|
> DoStoVoQ-SGD, d=8, $M=2^{12}$, **VGG16**  | $20\times$ | 92.1\%||
>
>
> $\ ^1$Compression factor is not directly reported in [Atomo]. To the best of our understanding, optimal results were obtained with an SVD with $s=3$, that would correspond to a compression factor of $\sim 6$.
>
> **Timing.** Thanks for raising this interesting point. Please refer to the comments for Reviewer WPUh for a discussion on the memory complexity.
> When evaluating the overall timing of our method, it is important to first highlight the fact that all experiments are performed **in a simulated environment**, that is on a single GPU on which users are simulated. Consequently, **we do not *actually* benefit from communication gains obtained by compression**.
>
> Regarding the relative overhead, it depends on the type of model/dataset used for training. We obtained the following results:
>
> CIFAR10, VGG16 | Time per epoch  | Compression factor|
> --- | ---:| ---:|
> No compression             | 30s |  1x  |
> QSGD                       | 51s | 8x |
> DoStoVoQ ($M=2^{10}$, d=8) | 52s | 25x  ||
>
> ImageNet, ResNet18 | Time per epoch  | Compression factor|
> --- | ---:| ---:|
> No compression             | 1925s |  1x  |
> QSGD                       | 2702s | 8x |
> DoStoVoQ ($M=2^{10}$, d=8) | 2523s | 25x  ||
>
>
> We thank the reviewer again for his/her numerous questions, and hope our answers clarified his concerns. We would be happy to provide more elements if there remain any unresolved questions.
>
> ---
>
> - [1] QSGD: Communication-Efficient SGD via Gradient Quantization and Encoding,  Alistarh et al.
> - [2] Mishchenko et al., 2019, "Distributed Learning with Compressed Gradient Differences"
> - [3] "Stochastic distributed learning with gradient quantization and variance reduction", S. Horvath, et al., 2019.
> - [4] vqSGD: Vector Quantized Stochastic Gradient Descent, Venkata Gandikota et al., 2020.
> - [5] "Federated Learning with Compression: Unified Analysis and Sharp Guarantees", Haddadpour et al., AISTATS 21.
> - [6] "Error Feedback Fixes SignSGD and other Gradient Compression Schemes", SP Karimireddy et al., 2019, ICML.
> - [7] The Error-Feedback Framework: Better Rates for SGD with Delayed Gradients and Compressed Communication, SU Stich et al., JMLR 2020
> - [8] Linearly Converging Error Compensated SGD, E Gorbunov et al., NeurIPS, 2020
> - [9] Xu H., et al., “Compressed communication for distributed deep learning: Survey and quantitative evaluation”

---

> > ### Comment · Reviewer_jQuP · 2021-08-22
> > **Thank you for the thorough response**
> >
> > I commend the authors for providing such a thorough response. Part of my concerns is addressed by the extra experimental results provided in the rebuttal.
> >
> > For the practicality of the proposed algorithm, my concern still remains. In my view, the final goal that a comm-efficient distributed training method should aim at is to reduce the end-to-end training time. For instance, to achieve 95% accuracy on CIFAR-10, it can be 3x faster than vanilla SGD. Otherwise, it will be really hard to tell the value of the proposed method.
> >
> > To achieve real end-to-end speedup, it is really important to focus on the extra encoding/decoding costs introduced over compression rate, as mentioned in [1]. By looking at the extra results provided in the rebuttal, it seems that DoStoVoQ has a high encoding cost. Is it caused by a less optimized implementation?
> >
> > One another question: Is DoStoVoQ compatible with the all-reduce operation used in real systems, e.g., NCCL. Since it is widely known that a gradient compression method that is not compatible with all-reduce is hard to scale in practice [2].
> >
> > I'm sorry to be so harsh on the practicality of the proposed algorithm, but as a researcher, I think we should pursue are theoretically grounded methods that can really be adopted in products and make a real-world impact.
> >
> > [1] https://arxiv.org/abs/2103.00543
> >
> > [2] https://arxiv.org/abs/1905.13727

---

> > > ### Author Response · Authors · 2021-09-01
> > > **Thank you for your response**
> > >
> > > Thank you for your remarks and positive comments. We agree that the practicality of algorithms is very important in machine learning, and we believe that our approach has the potential to be used in practice, and is also a first step towards a better understanding of some aspects of compression algorithms.
> > >
> > > First, we would like to stress that *the goal of compression is twofold*
> > > 1. Limit the number of bits exchanged: this is beneficial for the  bandwidth usage, energy consumption, etc. . This aspect is described as (one of) the main motivation of compression in [Kairouz et al, e.g. at pages 13, 32].
> > > 2. Accelerate the end to end process.
> > >
> > > In our manuscript, we mostly focused on the first aspect. We agree that providing end to end speed-up is also of great interest, but this is dependent on the communication setting and thus harder to evaluate. In our rebuttal, we showed that the encoding time of DoStoVoQ is similar to the one of QSGD. Both methods should be able to achieve similar speed-ups in an end-to-end implementation.
> > >
> > > Regarding the important encoding time of DoStoVoQ it is likely that it could be reduced by a better implementation of our current technique. Moreover, refined nearest neighbor encoding could also be considered, for example relying on approximate algorithms (e.g., k-d tree) with pre-computed approximations (that typically would require some adaptations, additional storage, and to use pseudo-random codebooks).
> > >
> > > Finally, regarding all-reduce, many methods, including QSGD, SignSGD, Atomo, Top-K are indeed not compatible with all-reduce. We agree that this aspect may be important in practice and we will add a reference to [1]. Yet, this is also slightly orthogonal to the main message of our approach: in order to design better compression algorithms, improving the understanding of the theoretical properties (distortion, bias, independence) and proposing new theoretically grounded compressors is also very important.
> > >
> > > We hope this clarifies our intent and approach. Thank you again for your constructive comments and consideration.
> > >
> > > ---
> > > - [1] https://arxiv.org/abs/2103.00543
> > > - [Kairouz et al] Advances and Open Problems in Federated Learning, Peter Kairouz et al., https://arxiv.org/pdf/1912.04977.pdf

---

### Official Review · Reviewer_WPUh · 2021-07-16

**Rating:** 7
**Confidence:** 3

**Summary:**

This paper proposes a new unbiased compression operator based on Voronoi vector quantization. Its theoretical properties are analyzed and the variance bound is proved. Then it is applied in the Federated Learning setting and compared to the previously proposed compressors.

**Limitations And Societal Impact:**

Limitations of the work were mostly addressed. Negative societal impact concerns are not applicable due to the theoretical nature of the paper.

**Main Review:**

Overall this is good work on vector quantization, which seems to be the first one to prove the applicability of this approach for such large models as deep neural networks. It presents a sound theoretical analysis and may give rise to further exciting research based on the proposed idea. This is mainly why I think it may deserve publication at the conference.

The submission is clearly written and well organized, though I believe it can benefit from moving some technical details on pages 4-5 to the Appendix.

My main concerns/requests are to explicitly and carefully say about the memory requirements and computational complexity of the proposed compression method and highlight the differences to the previous approaches. I would like the authors to measure the average time of the suggested technique required for compression and decompression. It is an important aspect, which in some cases limits the applicability of the compressors in the distributed learning [1, 2]. In addition, it is not compatible with All-Reduce, which significantly worsens the efficiency for the distributed training.

Besides, there is almost no experimental study of DoStoVoQ’s performance with respect to its algorithm parameters $L, M, P$. So, I hope that the current paper will be complemented with such results.

Minor issues/comments:

It seems that the authors missed a recent line of recent [3, 4] on compressors based on Kashin representation or Hadamard transform combined with stochastic quantization and the follow-up relevant papers.

[1] Xu H., et al. “Compressed communication for distributed deep learning: Survey and quantitative evaluation”. [Technical report](https://repository.kaust.edu.sa/handle/10754/662495), 2020.

[2] Agarwal S., et al. “On the Utility of Gradient Compression in Distributed Training Systems”. arXiv preprint arXiv:2103.00543, 2021.

[3] Caldas S., et al. “Expanding the Reach of Federated Learning by Reducing Client Resource Requirements”. arXiv preprint arXiv:1812.07210, 2018.

[4] Safaryan M., et al. “Uncertainty Principle for Communication Compression in Distributed and Federated Learning and the Search for an Optimal Compressor”. Information and Inference: A Journal of the IMA (2021).

[5] Saha R., et al. “Distributed Learning and Democratic Embeddings: Polynomial-Time Source Coding Schemes Can Achieve Minimax Lower Bounds for Distributed Gradient Descent under Communication Constraints”. arXiv preprint arXiv:2103.07578, 2021.

**Time Spent Reviewing:**

6

---

> ### Author Response · Authors · 2021-08-10
> **Response to Reviewer WPUh**
>
> We thank the reviewer for his/her time, careful reading, and insightful comments, and for finding that our approach can give rise to exciting research, and underlining the quality of the analysis and the writing.
>
> **Presentation and References.** Thank you for your suggestions regarding presentation and references. We will do our best to improve the presentation of the results page 4 and 5, by giving more intuitive interpretation of the results. We will also add the suggested references.
>
> **Memory and complexity:** This is an excellent point, that was also a concern of Reviewer jQuP. We will add a discussion on the memory, and time-complexity. Let us provide some elements here:
>
> - **Memory:** The only extra-memory requirement is the storage of the Codebook, of size $M\times d$, with (typically) $d=16$ and $M =4096$. This is often marginal w.r.t. the size of the model of size $D$.
> - **Time-Complexity:** At each step, on each node, the algorithm has to perform the following two steps:
>     - Loading a batch of data, computing backpropagation on this batch at the current model
>     - Performing compression.
>
> The balance between both obviously strongly depend on the time-complexity for backpropagation, that is highly dependent on the network architecture.
>
> For compression, their are two main steps:
> - Sampling the codebook. Experimentally, this is very fast.
> - Finding the nearest neighbors in the codebook for each bucket.
>
> Remark that this last step is extremely and easily parallelizable (it can be solved by performing a tensor product of two matrices).
>
>
> **Numerical values for time complexity.** Finally, we report overall timing. Here, it is important to first highlight the fact that all experiments are performed **in a simulated environment**, that is on a single GPU on which users are simulated. Consequently, **we do not *actually* benefit from communication gains obtained by compression**.
>
> Regarding the relative overhead, as explained above, it depends on the type of model/dataset used for training: we obtained the following results:
>
> CIFAR10, VGG16 | Time per epoch  | Compression factor|
> --- | ---:| ---:|
> No compression             | 30s |  1x  |
> QSGD                       | 51s | 8x |
> DoStoVoQ ($M=2^{10}$, d=8) | 52s | 25x  ||
>
> ImageNet, ResNet18 | Time per epoch  | Compression factor|
> --- | ---:| ---:|
> No compression             | 1925s |  1x  |
> QSGD                       | 2702s | 8x |
> DoStoVoQ ($M=2^{10}$, d=8) | 2523s | 25x  ||
>
>
>
> Regarding decompression, the central server has to generate the codebook and access its relevant elements. Both these steps are very fast.
>
>
> **Complementary results with respect to $M$, $L$ and $P$.**
> This is also a valid point. We will provide the accuracy as a function of $M$ and $L$ (or equivalently, $d =D/L$). We will also describe dependence on $P$.
>
> For example, for $d \in\{2,8,16\}$, we have the following accuracies on CIFAR10 (we increased the number of points until we consistently achieved more than 92%):
>
> CIFAR-10, VGG16 | d=2 | d=8  | d=16 |
> ---      | ---:     |  ---:| ---:|
> M= 32    |   92.2\%   |    91.7\%     |  90.9\%  |
> M= 128   |    92.4\%     |   91.8\%      |  91.8\%   |
> M= 256   |    -    |91.8%    |    91.9\%      |
> M= 1024  |    -     |     92.1\%    |    91.9\%      |
> M= 4096  |    -     |92.1\%   |    92.4\%       |
> M= 8192  |    -     |    -     | 92.0\%   ||
>
>
>
> We thank you again for your support and detailed comments, that will help improve the paper. We would be happy to provide more elements if there remain any unresolved questions.

---

> > ### Comment · Reviewer_WPUh · 2021-08-21
> > **Additional comments**
> >
> > I would like to thank the authors for their answers to my concerns and the additional experimental work they performed. I greatly agree with the chosen focus on *“a simple but reliable metric, the distortion on the vectors compressed”*. For me, it also seems like the best way to study such novel theoretical techniques.
> >
> > Although, when you mention *Federated Learning* in the title, it affects the expectations of people reading the work with respect to experimental results. A proper way to address it (in my opinion) would be to study the more or less real and practical federated learning setting, which involves protocols (datasets and problems) described e.g. in the paper [1]. In the current form, the word *“Distributed”* would be more suitable in the title, but it will impose more restrictions on the compressor to make it practically useful. These restrictions were described in recent works [2, 3].
> >
> > I had an additional look at the paper, and it forced me to double my point on reorganizing pages 4-5, so it will be easier for people working on practical compressors to follow it easier and maybe make the presentation “more compatible” with other works on gradient compression.
> >
> > I think that the practical contributions of the work can be enhanced by providing guidelines on optimally setting the parameters of the proposed compression method ($d, M$). Maybe you could include the simplified “recommended” asymptotical expressions for them and other parameters ($M, L, P$). This could be done in a simple setting of compressing random vectors without distributed training of neural networks.
> >
> > In addition, I have seen your response to Reviewer Td65 about the unbiasedness of the proposed method. If I understood correctly, the compressor is technically not unbiased and from the first glance, it seems as a serious issue because distributed methods with biased compressors can diverge [4]. Can you somehow verify the "unbiasedness" experimentally or show that the bias is indeed “tiny”? Please correct me if I am wrong, but for now, it seems that your method requires some sort of Error-Feedback [5, 6] mechanism to guarantee convergence?
> >
> > To conclude, I greatly encourage the authors to include a more detailed discussion of how the proposed method is compatible with the “systems” aspect with real federated (or distributed) learning settings [7]. It would be helpful to highlight the current limitations and provide ideas for further research.
> >
> > ***
> >
> > [1] Sashank R., et al. "Adaptive federated optimization." arXiv preprint arXiv:2003.00295 (2020)
> >
> > [2] Xu H., et al. “Compressed communication for distributed deep learning: Survey and quantitative evaluation”. [Technical report](https://repository.kaust.edu.sa/handle/10754/662495), 2020.
> >
> > [3] Agarwal S., et al. “On the Utility of Gradient Compression in Distributed Training Systems”. arXiv preprint arXiv:2103.00543, 2021.
> >
> > [4] Beznosikov A., et al. “On biased compression for distributed learning”. arXiv preprint arXiv:2103.00543, 2021.
> >
> > [5] Stich S. U. and Karimireddy S. P. "The error-feedback framework: Sgd with delayed gradients". Journal of Machine Learning Research, 21(237):1–36, 2020
> >
> > [6] Richtárik P., et al. "EF21: A new, simpler, theoretically better, and practically faster error feedback". arXiv preprint arXiv:2106.05203 (2021)
> >
> > [7] Kairouz P., McMahan H. B., et al. "Advances and Open Problems in Federated Learning". Foundations and Trends in Machine Learning Vol 4 Issue 1, 2021.

---

> > > ### Author Response · Authors · 2021-08-25
> > > **Response to additional comments**
> > >
> > > We thank you again for your comments and support. We will carefully take into account the modifications you suggested in the final version of the paper.
> > >
> > > Regarding the absence of bias, we would like to clarify two things:
> > > - the method described in the paper in Algorithm 1 and 2, in which the nearest codeword is rescaled by $(r_M(x))^{-1}$, **is unbiased** (theorem 1). Consequently, all guarantees *without error feedback* apply.
> > > - when *implementing* the method, we indeed rely on an estimation of $(r_M(x))$. However, as this function is very regular and only has to be estimated once, and the value can be computed with very high precision.
> > >
> > > For example, for $d=16$ and $M=2^{13}$ (one of the settings in Fig 1), $r_M(x)\in [0.5;1]$ and we  estimated  $r_M$ with 5 digits of precision (and this could easily be increased). In practice, it means that we sometimes rescale the vector by, say, $1.5754$ instead of $1.5753$. This does not impair convergence.
> > >
> > > We believe that this approximation should be considered as an implementation artifact, and that providing guarantees for the actual method (without estimation) is the most relevant approach here.
> > >
> > > We hope this answers your question on this point, and thank you again for your feedback.

---

> > > > ### Comment · Reviewer_WPUh · 2021-08-28
> > > > **Final acknowledgment**
> > > >
> > > > Thanks a lot for your clarifications. I agree with your remarks about $r_M(x)$ and now see that this is indeed an *implementation artifact*.
> > > >
> > > > I hope that the discussion was helpful and will beneficially affect the resulting paper.

---

### Official Review · Reviewer_Td65 · 2021-07-17

**Rating:** 6
**Confidence:** 3

**Summary:**

The paper presents a new unbiased gradient (vector) quantization method for communication efficient distributed optimization.
The authors establish bounds on expected quantization error and give convergence guarantees in convex optimization settings.

**Limitations And Societal Impact:**

No direct societal impact

**Main Review:**

There has been considerable interest in techniques for communication compression in distributed optimization. This is motivated by increase in size of datasets and models as well as new paradigms like Federated learning. Thus, the topic that the paper considers is important and of wide interest.

The paper proposes a new vector quantization method which is as follows: the method first samples a random codebook from some appropriate probability distribution. Then a vector is mapped to its nearest point in the codebook (hence the term voronoi quantization).
The paper deals with design of unbiased methods, however the aforementioned point has a directional bias. Therefore, finally, an appropriate scaling is done for debiasing. Hence, what is communicated is the index of the point in the codebook as well as the scaling factor.
The paper is mostly well-written. The authors present a thorough theoretical and empirical analysis.

**Biased vs unbiased:**: The authors motivate the design of unbiased methods by the fact that with unbiased compression, there is a benefit from averaging due to variance reduction.
There is a typo in the equation after line 96: the second term should be $K^{-1}E\Vert \text{Comp}_1(x) - E[\text{Comp}_1(x)]\Vert^2$.
The authors say that to correct the directional bias, theyneed to compute the quantity $r_M^p$, which is intractable, but can be approximated by Monte-carlo methods. This approximation error therefore contributes to a (small?) bias in the gradient estimation, hence the method is not *fully* unbiased.

**Empirical comparison with biased compression methods**:
The empirical results support the proposed method over other unbiased methods. However, I wonder how does it perform compared to biased compression methods, like top $k$?


**Improvement over scalar quantization?** The authors prove upper bounds on expected quantization error (Theorem 2). However, it seems to me that in order to get constant error, we need to set $M$ such such $\log M \approx d\log{d}$. Hence the communication size is $O(d\log d)$. Scalar quantization methods instead, for constant accuracy communicate $O(d)$ bits. Am I misunderstanding something here?

**DoStoVoQ: Impact of parameter $L$**: The only change in the second algorithm ``DoStoVoQ" is that it partitions the gradient vector into $L$ buckets and vector quantizes each. However, in the quantization error guarantee, there is no dependence on $L$? Why is it so? If there is indeed no dependence, why do we care about this algorithm at all?

**What if we repeat codebooks?**
In the method presented, a new codebook is sampled i.i.d. everytime a new vector is compressed. What is the issue if we repeat the same codebook, say among all workers, in an iteration?

**Convergence guarantee**: In line 316, the authors mention the convergence rate of DoStoVoQ-VR-DIANA (a variant in the appendix), however there is no mention of the the convergence rate of DoSToVoQ, the method in the main paper.

**Return communication compression?**:
In the algorithm, only the communication from workers to master is compressed. There is no mention of communication of updated model back to the workers.
Do the authors propose to compress the return communication too, or not? If yes, do the convergence guarantee still apply?

**Time Spent Reviewing:**

3

---

> ### Author Response · Authors · 2021-08-10
> **Response to Reviewer Td65**
>
> We thank the reviewer for his/her time, careful reading, and insightful comments.  We also thank the reviewer for underlining the thorough theoretical and empirical analysis, and for suggesting several extensions. We answer to each comment afterwards.
>
>
> **Biased vs unbiased** Thanks for pointing the typo!  We agree that, technically, the approximation resulting from the MC approximation would generate a tiny bias. However, as the function $r_M^p$ only depends on $M$ and $p$, which only have a few typical values, this function can be computed beforehand with a very large number of Monte Carlo samples. Moreover, its regularity (see Fig. 1 and the theoretical results in the Appendix), ensures the discretization does not badly impact its approximation.  Overall, the variance on the (Monte-Carlo) estimates of this function is negligible. Note that this variance is given as output of the code provided in Appendix.
>
>
> **Empirical comparison with biased compression methods** This is a good remark. Indeed, top-$k$ is known often to result in good performance in practice. Remark that we compared to Top-$k$ in Table 1, 2, 5-8 in terms of distortion. Keeping the same setting as throughout the paper (i.e., no tuning of parameters, we use the best parameters of SGD, no Error Feedback), we performed a complementary experiment this week for top-$k$ ($k$ chosen to achieve $8\times$ compression), achieving **91.2\%** of Accuracy, below our **92.1\%** for DoStoVoQ.
> *Remark:* At the request of Reviewer jQuP, we also added the comparison to Cross-Polytope method from [8], to which we had also compared in Tables 1, 2, 5-8.
>
>
> CIFAR10, VGG16 | test accuracy | compression factor|
> --- | ---:| ---:|
> No compression             | 91.9\%         | 1x     |
> Top-$k$                    | 91.2\%         | 8x     |
> Cross-Polytope             | 90.9\%         | 16x    |
> DoStoVoQ ($M=2^{12}$, d=8) | 92.1\%         | 20x    ||
>
> This experiment will be added to the paper.
>
> **Improvement over scalar quantization?**
> A scalar quantization communicating $O(d)$ bits (say, for example, 4 bits per coordinate), results in a constant error **per coordinate**, thus the  quadratic error  on a $d$-dimensional vector also scales as $d$, i.e., $\mathbb E [||x- VQ(x, C_M)||^2] \varpropto d$.
>
> On the other hand, our Theorem 2 shows that  with $\log_2(M) = d$ (i.e., $M^{2/d}=4$), we achieve an error  $\mathbb E [||x- VQ(x, C_M)||^2] \simeq \frac{d}{8\pi e} p^{2/d}(||x||)$, also of order d.
>
> Overall, the communication complexity is never worse for vector quantization, and even typically allows to gain a constant factor.
>
>
>
>
> **DoStoVoQ: interest of the algorithm w.r.t. StoVoQ.** We would like to point that another crucial difference between DoStoVoQ w.r.t. StoVoQ is the handling of random seeds, which is crucial to obtain independent codebooks (see point hereafter).
> Beyond that, DoStoVoQ indeed extends the methodology, thoroughly analyzed in moderate dimension $d$ (typically 16), to a larger dimension $D$ (typically $10^5$). This mostly consists in rescaling and splitting the gradient. We introduced this algorithm because
> 1. It is the one ultimately used in practice in all experiments on CIFAR10 and ImageNet;
> 2. The splitting and rescaling steps as well as the independence of codebooks are often overlooked but have major importance, at least from the theoretical standpoint.
> 3. It is the simplest example to illustrate how to incorporate our methodology into any FL algorithm
>
>
> **DoStoVoQ: Impact of parameter L.** In Theorem 3, the dependency in $L$ appears through the dependency on $D= L \times d$, and is hidden in the $O$. We did not made this dependency more explicit in the main text as the primary purpose of Theorem 4 is to prove DoStoVoQ satisfies all the required properties to apply convergence in the literature, and to exhibit the dependency on $M$.
>
>
> **What if we repeat codebooks?** This is a crucial aspect of the method, to be able to leverage the theory developed in the literature, that is unfortunately often not highlighted enough in the corresponding papers: all compression operators **must be independent**.
> - if the same codebooks were used by several users at the same iteration, one would lose the mutual independence of the compressions, which is necessary for the proofs (see e.g., [4,6,7]). Typically, the consequence would be to lose the good dependency on the number of users, which is one of our key focus. This corresponds to statement (i) in Theorem 4.
> - if a codebook used in the past by one user was used again at any further iteration, then the gradient would not be unbiased conditionally to the past (more formally, conditionally to the $\sigma$-algebra $\mathcal G_{k+1}$ such that all iterates from the past are $\mathcal G_{k+1}$ measurable.). This corresponds to statement (ii) in Theorem 4.
> These two aspects are always leveraged (often without proper highlight) in the proofs for unbiased compression operators.
>
> This is also why the way we handle seeds in DoStoVoQ is so important: we have to ensure that each codebook is generated in a completely independent way.
>
> Not that, in practice, this does not impair the speed of the algorithm, as sampling new codewords is extremely fast.
>
> We will expand the discussion on those points, to clarify the importance of each aspect.
>
>
> **Convergence guarantee.** This is a good remark: as our focus was mostly on the compression process, we chose to provide the simplest "complete" algorithm, DoStoVoQ-SGD (including seed handling, splitting, etc.). Yet,  we wanted to state the best convergence rate that can be achieved with our technique, that indeed relies on an algorithm adapted to heterogeneity (DoStoVoQ-VR-DIANA). We will add the convergence rate of DoStoVoQ-SGD.
>
> **Return communication compression?** This is another good point: most of the literature [1,6,2] has mainly focused on the uplink compression, that is generally considered more expensive [3, sec. 3.5]. Bi-directional compression can be tackled either naively [5], but requires algorithmic adaptations not to degrade the performance [4]. Adaptation of DoStoVoQ to that setting is straightforward.
>
>
> We hope those explanations will clarify your concerns. We thank you again for your support and detailed comments, that will help improve the paper.
> We would be happy to provide more elements if there remain any unresolved questions.
>
> ---
> - [1] QSGD: Communication-Efficient SGD via Gradient Quantization and Encoding,  Alistarh et al.
> - [2] Xu H., et al., “Compressed communication for distributed deep learning: Survey and quantitative evaluation”
> - [3] Advances and Open Problems in Federated Learning,  Peter Kairouz et al.
> - [4] Preserved central model for faster bidirectional compression in distributed settings, C Philippenko, A Dieuleveut, 2021
> - [5] A Double Residual Compression Algorithm for Efficient Distributed Learning, Liu et al., Aistats, 2020.
> - [6] Mishchenko et al., 2019, "Distributed Learning with Compressed Gradient Differences"
> - [7] Stochastic distributed learning with gradient quantization and variance reduction, S. Horvath, et al., 2019.
> - [8] vqSGD: Vector Quantized Stochastic Gradient Descent, Venkata Gandikota et al., 2020.

---

### Official Review · Reviewer_jsrJ · 2021-07-19

**Rating:** 6
**Confidence:** 3

**Summary:**

Contribution:
1. The paper proposed a new unbiased vector quantizer constructed from Voronoi quantization (which quantized a vector to its nearest neighbor in the codebook) and random codebook generation.
2. Theoretical properties of the quantization scheme are studied, including relative variance and asymptotic growth of relative variance w.r.t dimension and size of codebook.
3. The proposed quantization achieved superior performance compared with a few baseline methods on training neural nets.


**Limitations And Societal Impact:**

I missed the discussions on the limitations of the work.

**Main Review:**

Overall I think the paper is well-written and has a solid discussion on the theoretical properties of the proposed quantization scheme. But the experiments section can be improved.
Advantages:

1. Thorough analysis is given to relative variance and the asymptotic growth rate of it with the increase of dimension/size of codebook.
2. The proposed quantization is unbiased and has a lower error compared with HSQ methods (the main basline quantizers) both in theory and in experiments.

Disadvantages:

1.The comparison of different quantizers is only done well on training VGG on CIFAR-10. Similar comparison should be done on more dataset and/or network architectures.
2. For the ResNet on Imagenet experiment, only the performance of the proposed quantization method is reported and the accuracy is below 70%, though with a higher compression rate than QSGD, the accuracy lower than that of QSGD (which is around 75% in the original paper).

-----------------------------------------after rebuttal--------------------------------

Thanks for the detailed feedback and the new experiments. I encourage the authors to strengthen the experiments on imagenet but the current version looks good to me. I will keep my score.




**Time Spent Reviewing:**

2

---

> ### Author Response · Authors · 2021-08-10
> **Response to Reviewer jsrJ**
>
> First, we thank the reviewer for his/her time, careful reading, and comments. We are glad the reviewer  appreciated the thorough theoretical analysis and in of the relative variance increase. The reviewer mentioned that the experiment section could be improved, we are happy to provide some complementary results. We also refer to the general comment above, and to comments to Reviewer Td65, WPUh, and jQuP, for other complementary comparisons.
>
>
> **Extension to more architectures.** The code provided in Appendix can be easily tested on VGG nets, Resnets and Densenets without additional coding. During the rebuttal week, we performed an additional experiment: a single user was simulated on the CIFAR10 classification task with a Resnet18 instead of the VGG16 model. SGD method reaches $94.5\%$ while our Dostovoq method reaches $94.4\%$ with a $\times 20$ compression.
>
> CIFAR10, ResNet-18 | Test accuracy | Compression factor|
> --- | ---:| ---:|
> No compression             | 94.5\%         | 1x     |
> DoStoVoQ ($M=2^{12}$, d=8) | 94.4\%         | 20x      ||
>
>
>
> **Performance on ImageNet.** On ImageNet, our goal was **not** to achieve the baseline accuracy, but to show that we could achieve **much higher compression rates** while getting a "reasonable performance". Especially, we stress that we performed **no parameter tuning** for the DoStoVoQ run, and used exactly the same parameters than the ones **optimized for SGD**. Indeed, theoretical perspectives suggest that the optimal learning rate changes with compression. However, we have aimed at being as much conservative as possible, and tried to not optimize it for obtaining a fair comparison with competitive works. Improving upon those results would require to perform a more careful analysis of our gradient estimate with the "simple metric" that we have introduced. Yet, we believe this is beyond the scope of this paper and would require a specific analysis on its own.
>
>
> Similarly, our experiments are performed without adding Error Feedback (that resulted in a 0.6\% improvement on CIFAR10 when we added it this week).
> Remark that EF is widely used in practice for methods achieving the best performance [2]. We chose not not focus on this technique which is orthogonal to our contributions and only theoretically supported for biased compression operators.
>
> Finally, we would like to highlight that many papers (e.g, Power-SGD, Atomo, GradiVeq, ...)  proposing new compression techniques did not perform experiments on ImageNet.
>
> We thank the reviewer again for his comments and support. We would be happy to provide more elements if there remain any unresolved questions.
>
> ---
> - [2] Xu H., et al., “Compressed communication for distributed deep learning: Survey and quantitative evaluation”

---

### Author Response · Authors · 2021-08-10
**Main Comment**

We thank all the reviewers for their insightful comments on our method and results.
In the following, we provide detailed answers to the questions raised by the reviewers, and provide some complementary results or figures on the performance of our method.


Reviewers appreciated the quality of the analysis "thorough theoretical and empirical analysis" (Td65), "thorough analysis" (jsrJ) and underlined "it presents a sound theoretical analysis and may give rise to further exciting research based on the proposed idea." (WPUh), and all found that the paper was clearly written. They also highlighted the importance of communication efficiency,  "a promising research direction", and that  "scales of the experiments are large enough to provide valid evidence on the performance of DoStoVoQ" (jQuP).

Some reviewers asked for complementary figures or experiments. We performed several short new experiments during the rebuttal week, to strengthen the comparison to competing approaches. Especially,
- We ran the code with a different architecture (ResNet18) on CIFAR10.
- We performed an experiment with strong heterogeneity, using DoStoVoQ-DIANA (alg. 5 in our paper's Appendix), showing the robustness of our compression approach.
- We had provided a detailed comparison of the distortion of our methods compared to several other methods, including Cross-Polytope [1] and Top-$k$. To be thorough, we have added a comparison in terms of accuracy on CIFAR10 (VGG16). Our method significantly outperforms these two approaches (about 1\% difference).
- We reported computational time.
- We provided more accuracy results, underlining the impact of parameters $L$, $d$ and $M$.
- We ran an experiment with Error Feedback.
- We compared to results obtained in the ATOMO paper.


Note that all our experiments are performed in the same setting, that is using the optimal parameters for SGD for all methods, without any extra-tuning. Remark that all the code was provided at the time of submission and easily allows to replicate our results or extend them to other architectures, without additional coding.

We will add these experiments to the revised version of the manuscript, and carefully take into account all reviewers suggestions.

We agree that these experiments allow to reinforce the comparison to competing methods, yet also want to underline  one important message of our paper: what we propose and carefully analyze is a new compression technique, and we describe how it can be integrated within (any) Federated Learning algorithm. We thus focused on analyzing this compression, its properties, and to demonstrate that we can leverage many of the convergence guarantees obtained in the literature. To analyse numerically this compression technique, we focused on a simple but reliable metric, the distortion on the vectors compressed. We analyzed these distortions both for synthetic vectors, and for vectors sampled from gradients along the training of convex problems, or deep learning models. **We believe that distortion metrics (or other "elementary" metrics) have the potential to be more insightful  than "aggregated" metrics (performance of an advanced algorithm, combining memory and EF, after 100 epochs on a large dataset, with a specific batch size, learning rate decay, momentum parameter, etc.), that may not reflect the actual quality of the compression technique, or highly depend on the computational power used for tuning.**


We thank the reviewers again and would be happy to provide more elements if there remain any unresolved questions.

---
[1] Venkata Gandikota, Daniel Kane, Raj Kumar Maity, and Arya Mazumdar. vqsgd: Vector quantized stochastic gradient descent. Aistats, 2021.

---

### Decision · Program_Chairs · 2021-09-27

**Decision:**

Reject

**Comment:**

The paper proposes a new way to quantize SGD updates, and provides theoretical analysis and empirical validation in distributed settings (but not it seems in Federated settings where multiple updates happen at clients). The paper has several interesting contributions, but also needs a better comparison with the literature and connection to the FL setting (if that is what it is targeted for). I would encourage a resubmission to a subsequent venue with these fixes.